# Mechanical Activation and Cation Site Disorder in MgAl_2_O_4_

**DOI:** 10.3390/ma15186422

**Published:** 2022-09-16

**Authors:** Cole A. Corlett, Matthias D. Frontzek, Nina Obradovic, Jeremy L. Watts, William G. Fahrenholtz

**Affiliations:** 1Materials Science and Engineering Department, Missouri University of Science and Technology, Rolla, MO 65409, USA; 2Neutron Scattering Division, Oak Ridge National Laboratory, Oak Ridge, TN 37830, USA; 3Institute of Technical Sciences of SASA, Knez Mihailova 35/IV, 11000 Belgrade, Serbia

**Keywords:** spinel, neutron diffraction, inversion, mechanical activation

## Abstract

The synthesis and crystallographic site occupancy were investigated for MgAl_2_O_4_ with and without mechanical activation of the precursor powders. Heating to 1200 °C or higher resulted in the formation of a single spinel phase regardless of whether the powders were mechanically activated or not. Neutron diffraction analysis was used to determine cation site occupancy and revealed that mechanical activation resulted in a lower degree of cation site inversion compared to the nonactivated materials, which indicated that the powders were closer to thermodynamic equilibrium. This is the first study to characterize the effects of mechanical activation on crystallographic site occupancy in magnesium aluminate spinel using neutron diffraction.

## 1. Introduction

Magnesium aluminate spinel, MgAl_2_O_4_, is the only stable binary compound in the MgO–Al_2_O_3_ binary system at ambient pressure. It is the prototype for the spinel structure and is attractive due to its high melting point, corrosion resistance, mechanical properties, and low cost [1,2]. Many different routes have been used to synthesize spinel-based ceramics, including direct solid-state reactions of oxides, wet chemical precipitation, and mechanical activation (MA) [3,4,5,6,7,8,9,10]. For most applications, the choice of synthesis method is based on the desired particle size and purity of the spinel powder.

In naturally occurring MgAl_2_O_4_ spinel that achieved thermodynamic equilibrium over geologic timescales, Mg atoms (designated A cation herein) occupy one-eighth of the tetrahedral sites in the lattice [11]. Similarly, Al atoms (designated B cation herein) occupy one-half of the octahedral sites. Due to similar cation radii, Mg can occupy some octahedral sites and Al atoms can occupy some tetrahedral sites, a process that is called inversion. The inversion parameter (*i*) is the fraction of A lattice sites (i.e., Mg sites) that are occupied by Al atoms. The partially inverted structure can be represented as (A_1−*i*_B*_i_*)–[A*_i_*B_2−*i*_]Z_4_, where *i* is the inversion parameter, with tetrahedral site occupancy given in parentheses and octahedral site occupancy given in square brackets [12]. Site inversion in synthetic MgAl_2_O_4_ can range from 0.1 to 0.6, but powders synthesized by solid-state reaction typically have an inversion parameter that is around 0.25 [13]. Site inversion is difficult to detect using X-ray diffraction (XRD) due to the similar X-ray scattering cross-sections for Mg [12] and Al [13]. In contrast, the neutron scattering cross-sections for Mg (0.06 barn) and Al (0.23 barn) are significantly different, which enables the use of neutron powder diffraction (NPD) to determine site occupancy [14].

Mechanical activation is a high-energy ball milling process that induces physical and chemical changes in materials [6,15,16]. Mechanical activation reduces particle size and induces defects in materials that increase the chemical activity and potential energy of the activated material relative to equilibrium materials. This process can increase the speed of reactions and drastically decrease the reaction temperatures [17]. Mechanical activation can decrease processing time in that it can enable mixing and particle size reduction of constituents and, in some cases, solid-state reactions of a multicomponent system in a single process [18].

The aim of the present study was to determine the influence of mechanical activation on site occupancy in the MgAl_2_O_4_ spinel (MAS).

## 2. Experimental Procedure

Magnesium oxide powder (≥99%, −325 mesh, Sigma-Aldrich, St. Louis, MO, USA) was heated in air at 10 °C/min and calcined at 1000 °C for 4 h to decompose any hydroxide or carbonate species. The calcined MgO powder was then mixed with alumina powder (A16-SG, Almatis, Leetsdale, PA, USA) by dry ball milling for 24 h with ½” alumina medium using a medium-to-powder mass ratio of 1:1. After mixing, the precursor powder was ground using a pestle and mortar and sieved to 80 mesh. Approximately 12 g of the mixed powder was mechanically activated (MA) using high-energy ball milling (SPEX, Metuchen, NJ, USA; Model No. 8000) with 5 mm alumina spherical medium and an alumina mill jar. The medium-to-powder mass ratio was approximately 3:1. The powder was milled for 60 min using a cycle of 30 min on, 15 min off, and then the final 30 min to minimize powder heating. After milling, the powder was passed through an 80-mesh sieve with minimal grinding.

Powders that were either as-mixed or mixed plus mechanically activated were loaded into alumina crucibles and reacted at temperatures from 1200 °C to 1500 °C for 2 h in stagnant air (DT-30, Deltech, Denver, CO, USA). The heating rate was 10 °C/min, and the furnace was allowed to cool at its natural rate after the isothermal hold. Powders were denoted as either mixed (XD) or mechanically activated (MA) along with the calcination temperature (e.g., MA1200C denotes mechanically activated powder calcined at 1200 °C).

Reacted powders were lightly ground using a mortar and pestle and sieved to −200 mesh for X-ray diffraction (XRD; X’Pert Pro, PANalytical, Almelo, The Netherlands). Phase analysis of XRD data was performed by Rietveld refinement (RIQAS4, Materials Data Incorporated, Livermore, CA, USA). Phases were modeled using the appropriate ICSD data. Powder morphologies were examined using scanning electron microscopy (SEM; Raith eLine, Raith GmbH, Islandia, NY, USA). Powders were coated with a conductive Au/Pd layer before SEM.

Neutron diffraction patterns were collected on POWGEN at ORNL through the mail-in program [19]. The average sample mass was 1.5 g, which was loaded into vanadium sample cans. Collection time per sample was, on average, 3300 s for a proton charge of 4.5 × 10^12^ C. The central wavelength was 1.5 Å, which was used to assess lattice spacing from 0.485 Å to 13 Å. The total number of refined diffraction peaks was 133 per sample.

Rietveld refinement was performed using GSAS-II [20]. As a starting point for the refinement, the Crystallographic Information File (CIF 39161) from Yamanaka et al. was used assuming a site occupation of 100% Al and 0% Mg on 16c (octahedral position) and 100% Mg and 0% Al on 8b (tetrahedral position) [21]. The first step in the refinement was to scale calculations to the experimental values. Shape parameters were refined using a pseudo-Voigt with Gaussian and Lorentzian parameters. The Lorentzian parameters were not refined as sample effects were captured in the Gaussian portion. The background was described using a Chebyshev polynomial since it is not constant over the Q-range. For the unit cell, only the a-axis lattice parameters were refined, as no obvious reduction in symmetry occurred. The oxygen fractional coordinates and isotropic displacement were also refined. The second step of the refinement allowed the occupied site fractions for 16c and 8b to vary with the isotropic displacement parameter. The isotropic thermal motion parameters (U_iso_) on a specific site were constrained to be equal for Al and Mg. The specific site was constrained to be fully occupied (Al1 + Mg2 = 1, Al2 + Mg1 = 1). Both parameters are heavily correlated with Mg and Al occupancy. Furthermore, site occupancy could not be refined without these constraints or the restrains in the next paragraph.

The chemical composition target was 8 Mg and 16 Al per unit cell with a restrain weight factor of 150. If the initial assumption for the weight factor were too high, the refinement would stay at the initial configuration without changing the site occupancy. If the initial assumption for the weight factor were too low, then the chemical composition would deviate to become Mg-deficient and Al-rich. The weight factor was manually adjusted to restrict the nominal Mg content per unit cell to 8.00 ± 0.05 atoms for all refinements. The procedure described herein enabled optimization of all fitting parameters in the first step and then site occupancy in the second step. This refinement procedure was reproducible in contrast to sequential fitting methods that resulted in different site occupation with each iteration.

## 3. Results and Discussion

Heating the mixed or mechanically activated precursor powders to 1200 °C, 1300 °C, 1400 °C, and 1500 °C produced crystalline MgAl_2_O_4_. Analysis of the XRD patterns (Figure 1) for XD and MA powders confirmed that each specimen contained a single phase that indexed to MgAl_2_O_4_ (PDF: 01-074-1133) without any detectable residual MgO or Al_2_O_3_, or other impurity phases [22]. All of the calcining temperatures chosen for this study were able to fully convert the precursor powders to crystalline MgAl_2_O_4_.

As with the XRD results, the neutron diffraction patterns of both the XD and the MA materials indexed to the spinel structure. Figure 2 shows the neutron diffraction pattern of XD1500C, which is representative of the patterns collected for all of the materials in this study. Table 1 contains the peak listings of the main peaks indexed to the spinel structure. As seen in Figure 2, only peaks from a time of flight (TOF) of about 30,000 µs and larger are labeled with Miller indices. The labeled peaks were enough to show that the NPD pattern was consistent with spinel formation. The higher-order planes were too close to one another to label in this format but were included in the refinements. The most intense peak was at a TOF of roughly 32,255 µs, corresponding to the (440) plane. The next most intense peak was at approximately 45,631 µs, attributed to the (400) plane. The peak from the (222) plane was the third most intense peak at a TOF of about 52,692 µs. As with the XRD patterns, all of the NPD patterns were consistent with the formation of single-phase MgAl_2_O_4_.

The fractions of 16c and 8b sites occupied by Mg and Al were refined with isotropic displacement parameters for each powder. All 133 diffraction peaks that were observed for each specimen were used in each of the refinements. An impurity phase indicated by two small peaks was detected by NPD that was not detected by XRD. This impurity phase is likely residual MgO and/or Al_2_O_3_, and it was not included in the refinement. In Table 2, the results of the refinement are shown, where an R factor below 3% indicates a good refinement [23], and the Al1 and Mg2 columns correlate to the site occupancy of the 8b or tetrahedral position in the spinel structure. R is a figure of merit for Rietveld fitting [23]. According to the refinement, XD1500C was determined to have an inversion parameter of *i* = 0.13, indicating that 13% of the Mg atoms on octahedral sites were replaced by Al atoms. This was the lowest inversion parameter of the XD materials.

The Mg1 and Al2 columns correspond to the site occupancy of the 16c or octahedral position in the spinel structure. As stated above, the refinements were set such that each site was fully occupied, i.e., Al1 + Mg2 = 1 and Al2 + Mg1 = 1. The last column labeled “Mg” is the number of Mg atoms per unit cell after refinement, which was used as a final check of the validity of the fit. The weight factor was manually adjusted to keep this deviation ±0.05 from the nominal composition of eight Mg atoms per unit cell. The Rietveld refinement produced results with similarities with previous work on site occupancy and mechanical activation, as discussed in greater detail below.

The site occupancy data from Table 2 are summarized in Figure 3. Starting from the top, the XD1200C material returned the largest cation disorder or inversion parameter of 0.185. The next material studied, XD1300C, had an inversion parameter *i* = 0.149, significantly lower than the XD1200C sample. This trend of decreasing inversion value as calcination temperature increased continued, with XD1400C and XD1500C having inversion parameter values of 0.132 and 0.130, respectively. All of the mixed powders had inversion parameters of at least 0.13, with inversion decreasing as calcination temperature increased. The decrease in inversion parameter with increasing calcination temperature is thought to be due to the higher mobility of the cations at higher temperatures allowing the cations to shift to their equilibrium positions.

The degree of inversion of the mechanically activated powders did not show a clear trend with temperature. The inversion parameter was about 0.124 for MA1200C. The MA1300C and MA1400C samples each had lower values of cation site disorder with values of *i* = 0.075 for MA1300C and 0.074 for MA1400C. In contrast, increasing the calcining temperature to 1500 °C for MA1500C led to an increase in the inversion parameter to 0.097. As with the XD powders, the decrease in inversion parameter when calcination temperature increased from 1200 °C to 1400 °C was thought to be due to same mechanism that led to the decrease in inversion with increasing calcination temperature in the XD powders, i.e., increased cation mobility with increasing temperature. The increase in inversion parameter for MA1500C is thought to be due to a competing mechanism that increases cation disorder. While increasing temperature increases mobility and allows cations to move to their equilibrium positions, higher temperatures may also promote disorder through the increase in entropy and site disorder with increasing temperature.

Regardless of temperature, the inversion parameters for the MA materials were all lower than the lowest inversion value for the XD samples. Mechanical activation is known to produce defects in powder particles that result in a higher energy state compared to the nonactivated powders. Our hypothesis is that this higher-energy starting state allowed the MA powders to get closer to equilibrium than the XD powders for the same heat treatment temperature. These results generally agree with a previous study by Obradovic et al., who used Raman spectroscopy to characterize cation inversion in mechanically activated spinel powders [24]. Whereas Obradovic et al. found that inversion decreased with increasing temperature for activated materials, the present study showed the same trend for MA1200, MA1300, and MA1400, but a different mechanism appears to increase inversion for MA1500. The investigation by Obradovic et al. also found lower inversion for mechanically activated materials at some temperatures, while the present work revealed that all activated materials had lower degrees of inversion than the mixed powders. The differences in trends may be due to the measurement methods. Raman spectroscopy used by Obradovic et al. is more sensitive to the surface of powder particles compared to neutron powder diffraction, which penetrates through the powder particles and is a bulk measurement. Hence, the present study indicates that MA powders had lower inversion parameters than the XD powders after all heat treatments. In addition, both calcination temperature and mechanical activation influence on cation site disorder and, thus, the inversion parameter. Further investigation is necessary to determine if calcination temperatures above 1500 °C can produce material with greater cation site disorders than observed here.

## 4. Conclusions

The crystallographic site occupancies of stoichiometric MgAl_2_O_4_ powders were studied to determine the effect of temperature and mechanical activation on the cation site occupancy. Heating the various powders to at least 1200 °C produced single-phase MgAl_2_O_4_ spinel powders, with or without mechanical activation, which was confirmed by both XRD and neutron diffraction analysis. The mechanically activated MgAl_2_O_4_ powders had lower degrees of inversion than the lowest inversion parameter for mixed powders. The relationship between temperature and the degree of inversion indicated that competing mechanisms affect cation site occupancy. The lower inversion in MA powders compared to XD powders was attributed to the higher-energy state of the MA powders prior to calcination that provided a higher driving force to approach equilibrium (i.e., zero inversion) at the same calcining temperature. In addition, higher calcining temperatures appear to increase atom mobility and allow for lower degrees of inversion, but higher calcining temperatures can also result in more thermally activated site disorder. More research into how mechanical activation and temperature affect cation site disorder is needed to determine the mechanisms at play here.

## Figures and Tables

**Figure 1 materials-15-06422-f001:**
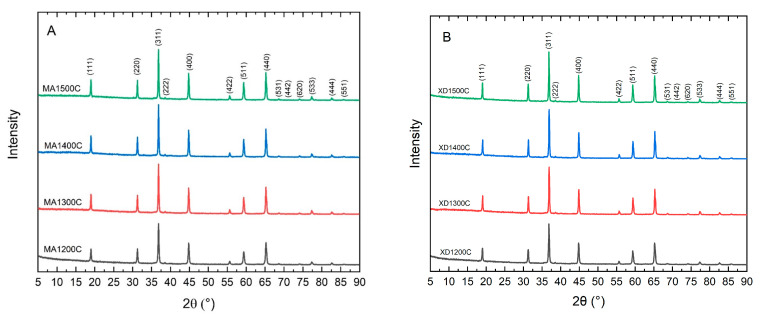
XRD patterns of the heat-treated MA (**A**) and XD (**B**) powders.

**Figure 2 materials-15-06422-f002:**
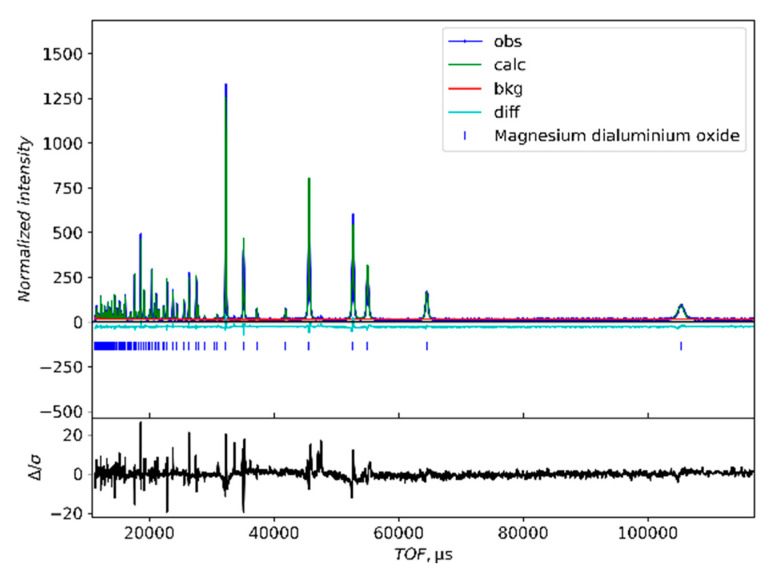
Representative neutron diffraction pattern for XD1500C with the Rietveld refinement fit. The black line is the residual (measured—calculated) plotted on a different scale than the pattern above).

**Figure 3 materials-15-06422-f003:**
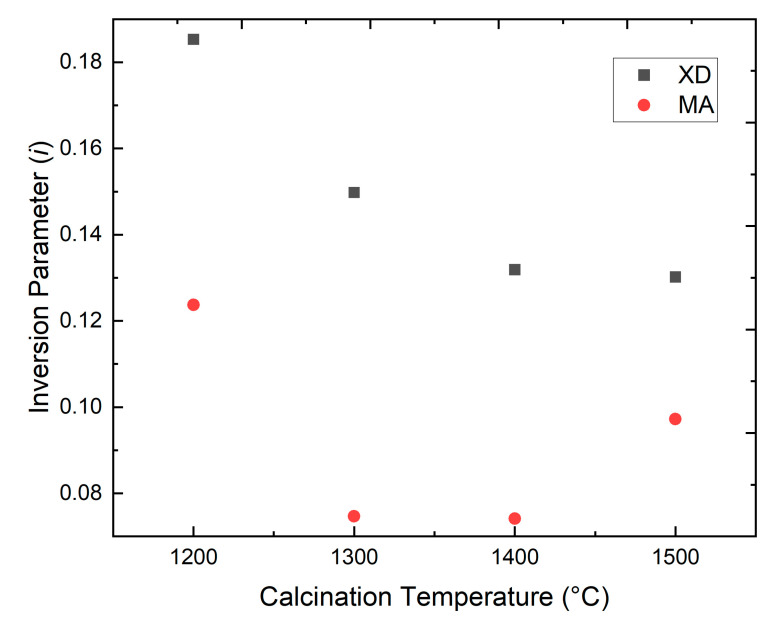
Inversion parameter vs. calcination temperature plotted for the spinel powders.

**Table 1 materials-15-06422-t001:** Peak listing of sample XD1500C displayed in Figure 2. RI is the relative intensity of the peaks.

TOF (μs)	d-Spacing (Å)	Miller Index	RI (%)
105,287.0	4.6677	*111*	7.27
64,554.0	2.8584	*220*	13.23
55,060.7	2.4376	*311*	23.81
52,692.5	2.3339	*222*	42.17
45,631.5	2.0212	*400*	58.99
41,924.3	1.8548	*331*	4.90
37,246.8	1.6503	*422*	6.49
35,135.6	1.5559	*511*	34.54
32,255.5	1.4292	*440*	100.00

**Table 2 materials-15-06422-t002:** The table shows the site occupation as a percentage. Al2 and Mg1 are on 16c, while Mg2 and Al1 are on 8b. Standard crystallographic cell Wyckoff sequence ecb.

Designation	Phase RF/RF^2^(%)	Al1 (%)	Mg2 (%)	Mg1 (%)	Al2 (%)	Mg	Inversion Parameter (*i*)
XD1200C	2.46/3.93	18.53	81.47	9.14	90.86	7.98	0.185
XD1300C	2.36/3.53	14.98	85.02	7.44	92.56	7.99	0.149
XD1400C	2.45/3.98	13.19	86.81	6.45	93.55	7.98	0.132
XD1500C	2.66/3.88	13.02	86.98	6.33	93.67	7.97	0.130
MA1200C	2.59/4.14	12.37	87.63	5.98	94.02	7.97	0.124
MA1300C	2.62/4.47	7.47	92.53	3.48	96.52	7.96	0.075
MA1400C	2.67/4.44	7.41	92.59	3.45	96.55	7.96	0.075
MA1500C	2.78/4.29	9.72	90.28	4.69	95.31	7.97	0.097

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
