# Peer review of "Mechanical Activation and Cation Site Disorder in MgAl2O4"

_materials, 2022, doi:10.3390/ma15186422_

Round 1

Reviewer 1 Report

In this work, the author determine cation site occupancy in magnesium aluminate spinel using neutron diffraction. However, there are still some issues to be addressed for improving this work.

1. The reference list should be arranged in the order of citation in the text.

2. What's R factor, it is better to give the definition of R factor in the main text.

3. What's “NMA materials” in the first paragraph in page 6.

4. How to define inversion parameter i and how to calculate the value of inversion parameter.

5.  How to understand that "The decrease in inversion parameter with increasing calcination temperature is thought to be due to the higher mobility of the cations at higher temperatures allowing the cations to shift to their equilibrium positions. " High mobility at higher temperature, which allows the cations to shift more easily, it seems to increase the inversion parameter.

6. The author consider that "While increasing temperature increases mobility and allows cations to move to their equilibrium positions, higher temperatures may also promote disorder through the increase in entropy and site disorder with increasing temperature. ". Further characterization is suggested to confirm the disorder.

7. The author consider that the differences in inversion may be due to the measurement methods between this work and reference 23. Depth discussions are suggested, why do surface sensitive measurement and bulk measurement can cause the different trends. What does it mean from aspect of materials.

Author Response

Dear reviewer, first of all, we would like to thank you for your comments. Secondly, we tried to answer to all of them, and we hope the paper will now be ready for publishing! Please, read our answers below:

Rev 1                                                                                         Comments and Suggestions for Authors

In this work, the author determine cation site occupancy in magnesium aluminate spinel using neutron diffraction. However, there are still some issues to be addressed for improving this work.

  1. The reference list should be arranged in the order of citation in the text. [Yes, you were right, ref. 7 was not in the right place, we went through all the references, and now they are right]
  2. What's R factor, it is better to give the definition of R factor in the main text. [R is a figure of merit for Rietveld fitting. The parameter is defined in Reference 23 in the paper. This is from Ref 22: “R F=(Σ hkl∣F O,hkl∣−∣F C,hkl∣)∕(Σ hkl∣F O,hkl∣) or based on F 2, R F 2=(Σ hklF O,hkl2F C,hkl2)∕(Σ hklF O,hkl2))” with the reference to Young: Young, R. A. (1993). “Introduction to the Rietveld method,” The Rietveld Method, edited by Young, R. A. (Oxford University Press, Oxford), pp. 1–38. We added the sentence in the text.]
  3. What's “NMA materials” in the first paragraph in page 6. [We change NMA to XD]
  4. How to define inversion parameter i and how to calculate the value of inversion parameter. [The inversion parameter (i) is the fraction of A lattice sites (i.e., Mg sites) that are occupied by Al atoms. This is clearly defined in the second paragraph of the Introduction, but we also added it in the text]
  5. How to understand that "The decrease in inversion parameter with increasing calcination temperature is thought to be due to the higher mobility of the cations at higher temperatures allowing the cations to shift to their equilibrium positions. " High mobility at higher temperature, which allows the cations to shift more easily, it seems to increase the inversion parameter. [At thermodynamic equilibrium, the inversion parameter should be zero. Heating results in a competition between the increased mobility that allows Mg atoms to move onto A sites in the lattice (i.e., reducing the inversion parameter) and thermally driven disorder that provides energy that promotes random site occupancy (i.e., increases the inversion parameter).  The statement here is intended to indicate that heating may enable Mg atoms to move to their equilibrium positions, thereby reducing the inversion parameter]
  6. The author consider that "While increasing temperature increases mobility and allows cations to move to their equilibrium positions, higher temperatures may also promote disorder through the increase in entropy and site disorder with increasing temperature. ". Further characterization is suggested to confirm the disorder. [We believe that the reviewer missed the point here. The Rietveld refinement of neutron diffraction data is the method being used to characterize site disorder. The statement is made based on the Rietveld refinement of neutron diffraction patterns that are presented in the text.]
  7. The author consider that the differences in inversion may be due to the measurement methods between this work and reference 23. Depth discussions are suggested, why surface sensitive measurement and bulk measurement can cause the different trends. What does it mean from aspect of materials? [Without direct comparison, we agree that neutron powder diffraction and Raman spectroscopy may have some differences in the absolute values of the inversion parameter. However, the analysis presented here is focused on the trends of the degree of inversion with different mechanical and thermal treatments, which we believe is a valid comparison.]

Reviewer 2 Report

The paper reviewed is devoted to the synthesis via mechanical activation pathway of spinel-based material and its characterization, including site occupancy factors’ refinement, using Rietveld technique. In general, the manuscript is nicely prepared, experimental data looks more-or-less reliable, but I still have several questions listed below that should be answered before the acceptance, including quite significant one. So, I believe that the paper can be published after major revision.

The major comment is related to methodology of the work. During refinement processes it was assumed the sites are fully occupied, which can be not so. I expect that at least two chemical analyzes (for MA and XD) should be additionally done. Probably EDX is the easiest one. I guess it should be rather easy to prepare parallel and polished pellets from your powder samples. If these experiments will give Mg:Al:O ratio equal to 1:2:4, than your further site occupancy factors’ refinement would be legal. Without chemical data all the discussed hypotheses are not so reliable and can’t be truly proved.

Minor comments:

- Corresponding author should be marked, and e-mails should be distributed between affiliations.

- Introduction: “…MgAl2O4, is the only binary compound in the MgOAl2O3 binary system.” – I think that it is not true. Of course, spinel is the main compound in the system, but it is not the only one. Even a short look through the ICSD reveals two more compounds MgAl26O40 (Jagodzinski, 1957) and Mg2Al2O5 (Enomoto et al., 2009). And PDF will likely give much more hints.

- Introduction and further: “…neutron powder diffraction (NPD) …” – I believe that this phrase should be re-written as powder neutron diffraction, since it describes the diffraction of neutrons (analogously to XRD – X-ray diffraction, which can be powder (P)XRD and single crystal (SC)XRD).

- Experimental: There is no need in sub-chapter “processing” if it is the only one, or some more sub-chapters should be added.

- Experimental: “… CIF from Yamanaka et al. was used …”. I recommend to add here also cif code (csd number), and also few words on why such an old file (1983) was used are definitely needed.

- Results: Reference for PDF 01-074-1133 should be given.

- Fig. 1a: Small peak before the 220 reflection at the MA1500C pattern is observed – what is it?

- Fig. 1: I recommend changing the Figure – too much free space between identical patterns. Shift them closer to each other. Besides, I recommend putting all 8 patterns together, since they are similar. Moreover, this will help to see the similarity much better.

- Fig. 2: What are the peaks that are slightly to the right from (400) reflection?

- Table 2: Short explanation of what is the last “Mg” column is should be given in caption or header.

- Table 2: I also recommend using similar numerical indices for the same site: Mg1 – Al1 and Mg2 – Al2.

- Table 2: It would be also useful to add a column with inversion parameters.

Author Response

Dear reviewer, we would like to thank you for useful comments, we hope out paper is now ready for acceptance! Please, read our comments bellow:

Rev 2                                                                                              Comments and Suggestions for Authors

The paper reviewed is devoted to the synthesis via mechanical activation pathway of spinel-based material and its characterization, including site occupancy factors’ refinement, using Rietveld technique. In general, the manuscript is nicely prepared, experimental data looks more-or-less reliable, but I still have several questions listed below that should be answered before the acceptance, including quite significant one. So, I believe that the paper can be published after major revision.

The major comment is related to methodology of the work. During refinement processes it was assumed the sites are fully occupied, which can be not so. I expect that at least two chemical analyzes (for MA and XD) should be additionally done. Probably EDX is the easiest one. I guess it should be rather easy to prepare parallel and polished pellets from your powder samples. If these experiments will give Mg:Al:O ratio equal to 1:2:4, than your further site occupancy factors’ refinement would be legal. Without chemical data all the discussed hypotheses are not so reliable and can’t be truly proved.

It is not clear why the reviewer believes that full site occupancy is not possible. We agree that some finite number of point defects will be present in any lattice. However, MgAl2O4 is a stable compound with a high melting temperature. The measurements in this case are all made at room temperature, so the number of point defects in the lattice will be considerably less than the inherent variability of EDS. Further, all of the materials in this study were made from a single batch of powder that was carefully weighed and then mixed.  Part of the batch was reserved in the as-mixed condition (XD compositions) and part underwent mechanical activation (MA compositions).  Even if site occupancy is not complete or the Mg:Al ratio is not exact, the two sets of powders have the same Mg:Al ratio based on the batching method. Because both Mg and Al have only one valence state, the amount of oxygen in the structure is fixed and does not vary under the processing conditions used, so the ratio of Mg and Al to O is not varied.  In addition to this reasoning, the refined lattice constant was 8.084Å, which is consistent with stoichiometric MgAl2O4. Spinel that is off-stoichiometry (i.e., the Mg to Al ratio is not 1 to 2) results in a linear reduction of the lattice constant (Ref: S.Lucchesi Zeitschrift für Kristallographie 209(1994) 714-719). From the refined lattice constant we can conclude that our samples were stochiometric. For comparison, a minimal off-stochiometric sample reported in Ref Leccessi has a= 8.04Å.

Minor comments:

- Corresponding author should be marked, and e-mails should be distributed between affiliations.[As suggested, that is done in the text]

- Introduction: “…MgAl2O4, is the only binary compound in the MgOAl2O3 binary system.” – I think that it is not true. Of course, spinel is the main compound in the system, but it is not the only one. Even a short look through the ICSD reveals two more compounds MgAl26O40 (Jagodzinski, 1957) and Mg2Al2O5 (Enomoto et al., 2009). And PDF will likely give much more hints.[The text was modified to say “MgAl2O4, is the only binary compound in the MgO-Al2O3 binary system at ambient pressure” which is true.  Other stoichiometries only form at extreme conditions.]

- Introduction and further: “…neutron powder diffraction (NPD) …” – I believe that this phrase should be re-written as powder neutron diffraction, since it describes the diffraction of neutrons (analogously to XRD – X-ray diffraction, which can be powder (P)XRD and single crystal (SC)XRD). [This is a case of semantics.  In ceramic science, the common terms are XRD, NPD.  If the editor suggests changing, we can change, but we have used the terms as they appear in the papers that we cite in this study.]

- Experimental: There is no need in sub-chapter “processing” if it is the only one, or some more sub-chapters should be added. [We removed the sub-section.]

- Experimental: “… CIF from Yamanaka et al. was used …”. I recommend to add here also cif code (csd number), and also few words on why such an old file (1983) was used are definitely needed.[This is a highly cited and reliable reference commonly used in the field.  No subsequent study has produced improved results, so this remains the definitive study in the field.] csd 39161

- Results: Reference for PDF 01-074-1133 should be given. [The powder diffraction file is a common database maintained by the International Centre for Diffraction Data. The reference can be the web site https://www.icdd.com/] So, ref 22 is added.

- Fig. 1a: Small peak before the 220 reflection at the MA1500C pattern is observed – what is it?  [We can *maybe* see something but it looks more like noise variance to us than a peak, it is right before or on the beginning of the upward movement on the (220) peak. Since this variance is not above the signal to noise ratio of the pattern, we do not believe that it needs to be identified or indexed.]

- Fig. 1: I recommend changing the Figure – too much free space between identical patterns. Shift them closer to each other. Besides, I recommend putting all 8 patterns together, since they are similar. Moreover, this will help to see the similarity much better.[We disagree with putting all 8 together. The spacing of patterns was made to prevent any overlap of the highest intensity peaks with the following pattern, so we disagree with this comment and did not make any changes.]

- Fig. 2: What are the peaks that are slightly to the right from (400) reflection? [We don’t see any peaks until (422)]

- Table 2: Short explanation of what is the last “Mg” column is, should be given in caption or header. [As stated in the text, “Mg” is the total number of Mg atoms per unit cell in the refinement.  The ideal number would be 8, so this is a check of accuracy. We added the phrase “Mg” is the number of Mg atoms per unit cell after refinement” to the caption to clarify this point]

- Table 2: I also recommend using similar numerical indices for the same site: Mg1 – Al1 and Mg2 – Al2. [The numerical indices are used to designate the 2 different sites for the atoms. So Mg1 is the site we believe it should be in and Mg2 is the inverted state. That means that Mg1 = Al2 and Mg2 = Al1. That is why the indices vary even though Mg1 and Al2 are the same site.]

- Table 2: It would be also useful to add a column with inversion parameters. [The inversion parameter values are in the paragraphs on page 10 after table 2, but we will list them]

Sample

Inversion Parameter (i)

XD1200C

0.185

XD1300C

0.149

XD1400C

0.132

XD1500C

0.130

MA1200C

0.124

MA1300C

0.075

MA1400C

0.075

MA1500C

0.097

Round 2

Reviewer 1 Report

1. What's "RI" in Table 1.

2. In the revised manuscript, the author defined that "The inversion parameter (i) is the fraction of A lattice sites (i.e., Mg sites) that are occupied by Al atoms. " It seems to be opposite to the result and discussion that "Based on the refinement, XD1500C was determined to have an inversion parameter of i = 0.13, indicating that 13 % of the tetrahedral sites were occupied by Mg atoms. ".  

Author Response

Reviewer 1

Comments and Suggestions for Authors

  1. What's "RI" in Table 1.

RI in Table 1 is “Relative Intensity”.  This term is now defined in the title of the table.

[ the sentence “RI is the relative intensity of the peaks” was added to the table title.]

  1. In the revised manuscript, the author defined that "The inversion parameter (i) is the fraction of A lattice sites (i.e., Mg sites) that are occupied by Al atoms. " It seems to be opposite to the result and discussion that "Based on the refinement, XD1500C was determined to have an inversion parameter of i = 0.13, indicating that 13 % of the tetrahedral sites were occupied by Mg atoms. ".  

The wording has been changed to be consistent between the definition of the inversion parameter and the Results and Discussion.

[the sentence in the Results and Discussion “Based on the refinement, XD1500C was determined to have an inversion parameter of i = 0.13, indicating that 13 % of the Mg atoms on octahedral sites had been replaced by Al atoms.” was updated]

Author Response

Thank you for clarification of the locations of the peaks.  The response to your inquiry has several parts.  First, the “peaks” that you have pointed out do not pass the test for statistical significance as defined on Pages 298-299, the book Introduction to X-Ray Powder Diffractometry by R. Jenkins and R.L. Snyder, which is that the intensity must be more than 2.5 times the standard deviation of the baseline intensity.  For example, the small peak at about 35.1° two theta in figure 1a that was just before the (311) peak of spinel has a maximum intensity of 250.4 with the average background intensity of 193.6 and a standard deviation in intensity of 24.9 over the ranges of 35 to 36 degrees two theta.  Based on these numbers, the peak is only about 2.3 standard deviations above background and is not considered statistically significant.  Similar analyses were done for the peaks in the other patterns.  Because the peaks are not statistically significant, they could be the result of random variations in the background intensity in the patterns.

Further analysis of the positions reveals different possible origins for the two peaks.  For the peak near 35.1 degrees two theta, this corresponds to the (104) peak for a-Al2O3.  For pure a-Al2O3, this peak has a relative intensity of 94%, so it is among the strongest peaks for alumina, which could indicate a small fraction of unreacted alumina remained in the spinel samples.  This peak was not apparent in the neutron diffraction data and should not have affected the fits for site occupancy that are the core of the paper.

The other peak at about 30.1 degrees two theta is more difficult to index.  It does not correspond to any peaks for a-Al2O3, MgO, or spinel.  Although it does not correspond to allowed peaks that are described on the standard patterns for these substances, the peak at 30.1° can be indexed to the (110) peak in MgO.  This is not an allowed peak for the rock salt structure, so it should have zero intensity.  This peak is also similar in position to the (112) peak for tetragonal zirconia, which is the most intense peak for this phase.  Hence, the most likely origin of this peak is from a small volume fraction if impurities introduced during the high energy ball milling step.  This impurity is not considered significant because:  1) the peak is low intensity and did not have a significant effect on the fitting; and 2) ZrO2 is not soluble in spinel and would not shift the position or relative intensity of the peaks used for the fits used to determine site occupancy.

If the editor recommends, we can add a short statement to the text of the paper. Because the peaks do not meet the criterion for statistical significance, we don’t believe that this addition is necessary.  However, we will modify the text, if requested by the Editor.

Round 3

Reviewer 2 Report

I'd like to thank authors for the work they have done. These data definitely should appear in the manuscript. Afterwards it can be accepted.